# Noise Reduction and Localization Accuracy in a Mobile Magnetoencephalography System

**DOI:** 10.3390/s24113503

**Published:** 2024-05-29

**Authors:** Timothy Bardouille, Vanessa Smith, Elias Vajda, Carson Drake Leslie, Niall Holmes

**Affiliations:** 1Department of Physics and Atmospheric Science, Dalhousie University, Halifax, NS B3H 4R2, Canada; vanessa.smith@dal.ca (V.S.); el676365@dal.ca (E.V.); carsonl@dal.ca (C.D.L.); 2Sir Peter Mansfield Imaging Centre, School of Physics and Astronomy, University of Nottingham, University Park, Nottingham NG7 2RD, UK; niall.holmes@nottingham.ac.uk; 3Cerca Magnetics Limited, Units 7–8 Castlebridge Office Village, Kirtley Drive, Nottingham NG7 1LD, UK

**Keywords:** optically pumped magnetometers, cylindrical shield, localization accuracy, phantom

## Abstract

Magnetoencephalography (MEG) non-invasively provides important information about human brain electrophysiology. The growing use of optically pumped magnetometers (OPM) for MEG, as opposed to fixed arrays of cryogenic sensors, has opened the door for innovation in system design and use cases. For example, cryogenic MEG systems are housed in large, shielded rooms to provide sufficient space for the system dewar. Here, we investigate the performance of OPM recordings inside of a cylindrical shield with a 1 × 2 m^2^ footprint. The efficacy of shielding was measured in terms of field attenuation and isotropy, and the value of post hoc noise reduction algorithms was also investigated. Localization accuracy was quantified for 104 OPM sensors mounted on a fixed helmet array based on simulations and recordings from a bespoke current dipole phantom. Passive shielding attenuated the vector field magnitude to 50.0 nT at direct current (DC), to 16.7 pT/√Hz at power line, and to 71 fT/√Hz (median) in the 10–200 Hz range. Post hoc noise reduction provided an additional 5–15 dB attenuation. Substantial field isotropy remained in the volume encompassing the sensor array. The consistency of the isotropy over months suggests that a field nulling solution could be readily applied. A current dipole phantom generating source activity at an appropriate magnitude for the human brain generated field fluctuations on the order of 0.5–1 pT. Phantom signals were localized with 3 mm localization accuracy, and no significant bias in localization was observed, which is in line with performance for cryogenic and OPM MEG systems. This validation of the performance of a small footprint MEG system opens the door for lower-cost MEG installations in terms of raw materials and facility space, as well as mobile imaging systems (e.g., truck-based). Such implementations are relevant for global adoption of MEG outside of highly resourced research and clinical institutions.

## 1. Introduction

Non-invasive neuromagnetic recordings with magnetoencephalography (MEG) sensor arrays provide an important window into the neurophysiology of the human brain. From a research perspective, MEG has informed our understanding of how neurophysiological signals change with factors such as task performance, age, and disease. From a clinical perspective, MEG has become a critical component of pre-surgical planning for patients with epilepsy, by providing non-redundant information to localize both epileptiform activity and areas of eloquence [1]. Since the 1960s, MEG sensors have used superconducting quantum interference device (SQUID) technology, developing from single-sensor recordings to commercially available whole-head high-density sensor arrays. Recent advances in the miniaturization of quantum sensors have seen the increased use of optically pumped magnetometer (OPM) technology for MEG recordings [2]. Studies of OPM MEG recordings indicate equivalent performance, as compared to SQUID MEG [3]. Other benefits include eliminating the need for cryogenics, flexible sensor placement [4,5,6], and enabling participant movement during recordings [7,8,9]. However, OPMs operate around a zero magnetic field resonance [10], placing even stricter requirements on the magnetic field in which they operate than SQUIDs.

The Biosignal Lab at Dalhousie University (Halifax, NS) is home to a mobile OPM MEG system. In brief, the OPM MEG system consists of a dual-wall cylindrical mu-metal shield mounted on a moveable cradle. The shield houses the OPM sensors in a 107-slot helmet-shaped sensor array mounted on a retractable participant support bed. Sensors are controlled by, and send data to, an electronics chassis outside of the shield and interfaced to a computer. During use, the cradle is placed on dampening materials to isolate the MEG system from building vibrations. This system design is in stark contrast to conventional (cryogenic and OPM-based) MEG systems which typically operate inside a large magnetically shielded room (MSR). Our approach reduces material and space costs associated with housing the imaging technology. Indeed, this MEG system design means that not only are the sensors and shield transportable (e.g., by a truck with sufficient load capacity, or wheeled between rooms in the same building) but the cost and space reduction of such a design could help more institutions acquire MEG technology. However, given the rarity of MEG systems in shielded cylinders [11,12,13,14] compared to MSRs, there is a need to quantify system performance.

Localization accuracy is a ubiquitous quality assurance metric for MEG systems, which is generally measured experimentally using a current dipole phantom. The current dipole is a vector quantity that represents current flow over distance, in the limit as distance approaches zero, which underlies most modern source estimation techniques in MEG [15]. Experimentally, a current dipole can be approximated by a current propagating over a few millimetres. An apparatus that generates current on dipole-like wire segments at precisely known positions and orientations is called a “phantom” and can be used to measure localization accuracy. MEG measurements are made as the wire segments are activated, and the calculated and known positions are compared to establish variability and bias. Previous phantom studies established the localization accuracy of MEG recordings at 1–5 mm with no systematic bias, using a range of phantom designs and current dipole positions and orientations [5,16,17,18,19]. However, phantom design, fabrication and calibration is challenging. As such, phantoms that can be easily integrated into OPM hardware and electronics are not readily available. Thus, there are few reports on the phantom-based localization accuracy of OPM systems.

The objective of this paper is to quantify the performance of the cylindrical shield system for MEG recordings. We examine the efficacy of the magnetic shielding in terms of magnetic noise attenuation and residual field isotropy. We expected that the shielding would sufficiently attenuate the environmental magnetic fields to allow the OPM sensors to remain stable. Additionally, we compare the field isotropy and noise power spectrum to similar reports in the existing literature. We also quantify localization accuracy based on simulations and current dipole phantom recordings with our OPM sensor array. Evidence from this study will validate the use of OPM MEG in a cylindrical shield for future studies and provide further evidence for the value of switching from SQUID to OPM sensors for MEG recordings.

## 2. Materials and Methods

Magnetic shielding is critical to attenuate noise due to the environment at the sensor array and to enable the proper functioning of OPM sensors. Figure 1a is a schematic of our cylindrical shielding solution. The cylindrical shield consists of inner and outer layers of mu-metal, with 43 cm and 48 cm radii and axial lengths of 1.9 m and 2.1 m, respectively (American Magnetics, Oak Ridge, TN, USA). The open end of the cylinder (for participant insertion) includes a 45 cm long reducer to a radius of 38 cm. The opposite ends of the inner and outer shields are closed with friction-fitted end caps. The thickness of all the mu-metal components is 1.8 mm. A smaller-diameter polymer cylinder, labelled Active Shield, includes coils for nulling the homogenous field along each cardinal axis (not used in this study).

The participant support is a retractable wooden bed on a non-magnetic rail system. The top of the bed is approximately 10 cm below the centre axis of the cylinder. A helmet-shaped sensor array is mounted to the head end of the bed at a fixed position (see Figure 1a for helmet location with respect to the shield), and the OPM sensors slide into slots in the helmet. Thus, the OPM sensor positions with respect to the helmet can be precisely known by measuring the depth of the sensor in each slot. The orientation is precisely defined for each slot based on the helmet manufacture, independent of depth. Sixteen Fieldline v2 OPM sensors with less than 20 fT/√Hz sensitivity (Fieldline Inc., Boulder, CO, USA) were available for this study. Device control and 24-bit data acquisition were managed via vendor-supplied v2 electronics chassis and a personal computer running vendor-supplied acquisition software (Recorder, v1.6.7) on the Windows 10 operating system.

Field measurements were made inside and outside of the cylindrical shield to estimate change for the DC and time-varying fields. All measurements of the magnetic field outside of the OPM system shielded cylinder were made using a Bartington MAG-13MSL 3-axis fluxgate magnetometer (Bartington Instruments, Witney, UK) with 6 pT/√Hz sensitivity connected to a National Instruments (NI, Austin, TX, USA) 16-bit USB-6216 multifunction I/O device via the Bartington PSU1 power supply. Data were captured to a Windows 10 laptop running custom Python software (NI-DAQmx v.0.5.7). For each measurement described below, data for all three axes were acquired concurrently for 10 s at 2000 Hz per axis and converted to magnetic field based on a conversion factor of 10 μT/V. The magnetometer was mounted on a wooden stand to vertically align with the centre axis of the shield. Field measurements were made at four locations around the shielded cylinder. At each location, the magnetometer was aligned to an outer corner of the shield and moved approximately 50 cm away from the shield. Two datasets were acquired—with the Bartington in DC-coupled and AC-coupled modes. DC-coupled recordings (acquired with USB-6216 analog input range set to ±5 V) provided sufficient range to capture the DC field strength. AC-coupled recordings (analog input set to ±0.2 V) provided increased sensitivity to small fields at high frequencies. DC-coupled magnetometer data were acquired parallel and anti-parallel to each axis. The DC magnetic field for each of the six orientations (three axes recorded parallel/anti-parallel) was calculated by taking the mean magnetic field strength in the 10-s recording. A final estimate of DC field strength along each axis was determined by averaging the parallel and anti-parallel values, to mitigate any directional bias in the magnetometer. AC-coupled recordings were used to estimate field change at frequencies above DC. 

All measurements inside the shield were made using three orthogonally oriented OPM sensors placed near the origin of the helmet array. The sensing volume of each OPM was placed more than 3 cm from the other sensors to minimize cross-talk between sensors [20]. DC magnetic field values along the radial axis of each sensor were captured directly from the acquisition software during the sensor configuration process. Bi-directional measurements were not made with the OPM sensors because the vendors reported no significant directional bias in these sensors. Time-varying fields were captured via 60 s recordings at 1000 Hz from all three OPM sensors concurrently. The OPM sensors were not used outside of the shield due to their limited operating range (DC field less than 100 nT magnitude). The Bartington fluxgate was not used inside of the shield due to its limited sensitivity. We acknowledge that using different sensors and acquisition paths limits accuracy in comparing fields.

For analysis of time-varying fields, a 200 Hz low-pass filter was applied to data followed by downsampling to 500 Hz. The amplitude spectrum was then determined as the square root of Welch’s method for power spectral density (scipy.signal.welch, SciPy v.1.5.2) with a 4 s window and 2 s overlap. Importantly, spectral analysis was aligned between OPM and fluxgate datasets to ensure equivalence in the amplitude spectra scales. The power line field strength was calculated as the magnitude of the 60 Hz peak in the magnetic field amplitude spectrum. For data inside and outside of the shield, field components along each cardinal axis were combined to determine the field vector magnitude. For each location outside of the shield, field attenuation, at DC and all frequencies, was estimated as a log-ratio of field power (i.e., field vector magnitude squared) inside and outside of the shield and reported in decibels. A best estimate across the four locations was determined as the average of the four field difference values. While a more accurate estimate of field change could be achieved by measuring the field at the helmet location without shielding, space and technical challenges limited our ability to acquire these data. However, the consistency of the field vector magnitude at the four locations outside of the shield suggests that these results would be consistent. Further, shielding factors can be definitively determined by applying a known time-varying field outside of the shield and measuring this specific signal inside and outside of the shield. However, this approach was beyond the scope of this paper.

To better characterize the spatial distribution of the residual field inside the shield, we determined the DC magnetic field vectors at regular grid locations in a rectangular volume of interest encompassing the OPM helmet array position. This is important information to quantify the isotropy within the volume that houses the sensor array. An isotropic environment, with maximal field gradients on the order of 50 pT/cm, reduces artefact signals due to sensor movement, which mitigates vibration artifacts and enables on-head MEG recordings without restricting head movements [6]. Both the inner and outer shield were degaussed prior to field mapping.

The helmet was removed for this process. A wooden plate with 3D printed sensor mounts was designed and built, such that sensors could be mounted in all three Euclidean orientations at reproducible positions within the shield. OPM sensors were arranged on the plate in a 4 × 4 rectangular grid in the x–y plane (i.e., vertical and participant left–right), with 8 cm spacing between sensors. The plate was then moved along the *z*-axis to collect DC field data at 4 cm intervals covering 24 cm (see Figure 1a and locations with respect to the shield). The acquisition procedure was repeated for each Euclidean orientation. Thus, 128 field vectors were captured over a 24 × 24 × 24 cm^3^ volume encompassing the helmet. For each sensor and orientation, DC field values were captured from the vendor-supplied acquisition software during the sensor configuration process. Time-varying recordings inside the shield indicate magnetic field changes less than 100 pT over 3 min (see Figure 1e) in a DC field of magnitude 10–40 nT. Thus, we were confident that the field was sufficiently stable to allow us to accurately map using consecutive measurements at multiple positions and orientation. Vector field mapping was repeated over days and weeks to assess the long-term stability of the residual field, with no further shield degaussing performed during this interval. The spatial distribution of the vector field was quantified based on the coefficients resulting from a spherical harmonic expansion up to fourth order [21,22]. See Table A1 for the spherical harmonic expansion equations. 

The accuracy of the spherical harmonic modelling of the experimental data was confirmed using a train–test framework. An additional field map was captured using intervals of 2 cm in the *z*-axis, as opposed to 4 cm. Data captured on the original 4 cm z-spacing were used to generate spherical harmonic coefficients (i.e., training set). These coefficients were then used to predict field vectors at locations on the additional z-locations withheld from model generation (i.e., test set). Model fit was assessed as the correlation between the predicted and measured fields along each Euclidean orientation for the test set. A high correlation here confirmed that our spatial sampling was sufficient to represent the field map as a spherical harmonic expansion.

To quantify localization accuracy of the MEG system, measurements were made using a custom-built current dipole phantom. We designed a “dry phantom” [17], which uses wire triangles to generate short tangential current segments at known positions and orientations. Figure 1b,c show the phantom, which consists of a platform and clamps, an arc, and dipole clips. The clamps are used to secure the platform to the helmet and align it with the helmet coordinate frame. Within the helmet, the x-axis passes through the centre of the C9 and C10 sensor locations, and the *y*-axis passes through the Nz sensor location (see Figure 1b). The phantom platform sits in the x–y plane with its centre at the origin. The arc is secured orthogonally to the platform at one of six azimuthal angles to define a 6.5 cm radius semicircle around the platform origin. The 3 mm wide dipole clips are then secured to the arc at known polar angles. A current dipole is created by running a wire from the origin over the dipole clip and back to the origin. The radial wire segments generate an insignificant radial magnetic field at the sensors because they are close to anti-parallel and approximately radial to the closest sensors. Thus, the 3 mm tangential segment (under the dipole clip) is the main source of the measured radial field at the OPM sensors. Four current dipoles were installed on the arc for this experiment at polar angles of 65, 25, −15, and −55 degrees. Data were captured at azimuthal angles of 0, 60 and 120 degrees.

During data acquisition, the circuit for each current dipole was activated (consecutively) by 100 half-cycles of a 5 Hz cosine wave at 2.5 V, with a 200 ms inter-stimulus interval. This signal was passed through a 500 kΩ resistor to generate a 5 μA current over the 3 mm wire segment (i.e., approximately 15 nAm current dipole peak amplitude). Sixteen channels of OPM data were collected continuously at 1000 Hz sample rate during dipole activation. For each recording, three OPM sensors were placed approximately 30 cm away from the helmet (further into the shield) and orthogonally to each other to act as a reference array. The reference array channels were spaced roughly 10 cm apart to ensure cross-talk was not a substantial contributor to the reference signals [6]. The remaining sensors were placed in helmet slots and inserted such that 50.5 mm of the sleeve was visible outside of the helmet. At this depth setting, the OPM sensitive volume is roughly at the inner surface of the helmet array. To capture the full spatial topography associated with each current dipole, we completed multiple consecutive OPM recordings, where the sensor array was shifted to different recording sites on the helmet for each recording. The reference array was also captured for each recording. Thus, we were able to record OPM data from a maximum of 104 of the 107 locations on the helmet array. Three locations were unavailable for recording because they were used to mount the phantom (Nz, C9, C10). Some sensors were excluded due to configuration issues, resulting in between 89 and 104 recording sites per dataset. 

MEG data were analysed in Python using the development version of MNE (v1.7.0) and sklearn (v1.0.2) software libraries. For each OPM channel (sensors and references), data were band-pass filtered (1 to 20 Hz), baseline-corrected and detrended via linear regression against time, and windowed for edge effects using a 2-s cosine window at the start and end of the recording. Data for the sensor array were updated to include the position and orientation of the channel in the helmet. Following these steps, reference array regression (RAR) was completed for each sensor in each recording. Linear regression was performed, with each reference signal as a dependent variable and the sensor signal as the independent variable. This resulted in one intercept and three coefficients that “minimize the residual sum of squares between the observed targets in the dataset, and the targets predicted by the linear approximation” (scikit-learn.org). The modelled data were then subtracted from the sensor signal to generate a new residual magnetic field time course for each OPM in the sensor array. As a final noise reduction step, homogenous field correction (HFC) was applied to attenuate magnetic fields with a consistent and homogenous spatial pattern over time [23]. 

For each recording, event markers were added to the data based on the timing of peaks in the current dipole driver signal. RAR-analysed OPM data were separated into 400 ms epochs centred on the driver signal peak times. Epochs were averaged in consecutive bins of 100 trials to generate four evoked fields (one for each current dipole), and evoked fields were baseline-corrected based on data occurring in the 50 ms interval prior to dipole activation. Reference channels were separated into a second evoked field dataset at this stage, to ensure insensitivity to the current dipoles. Finally, evoked fields from each recording were combined to generate whole-head evoked field data for each current dipole. Dipole fitting was completed using the mne.fit_dipole function applied at time zero. The identity matrix was used for covariance since the full covariance matrix cannot be calculated due to the use of concurrent recordings. The vector displacement between known and measured current dipole position was the primary outcome per dipole. Displacement vectors were analysed across all activated dipoles to assess variability and bias in localization (i.e., localization accuracy). Variability along each axis was defined as the standard deviation associated with the displacement vector’s projection to that axis across dipoles. Bias was defined as the ratio between the mean displacement vector along that axis and the variability. 

For broader investigation of the effectiveness of our noise reduction techniques, we also completed preprocessing steps on wide-band (0.5–100 Hz plus power line notch filter) OPM data. Baseline correction, detrending and windowing were still applied in this case. Amplitude spectra were made to compare signal quality following various noise reduction steps. As with shield performance, attenuation due to data analysis as a function of frequency was quantified in decibels. 

Given the experimental nature of our phantom, we completed simulation studies to compare localization accuracy. Current dipoles were simulated with the same radius as in the experimental recordings, at varying azimuthal and polar angles. It was found that a simulated current dipole magnitude of 25 nAm resulted in evoked fields of similar magnitude to the experimental data. As such, all simulations were completed with this dipole strength. OPM data were calculated based on current dipole activations at the source location with Gaussian noise added over time. Data were low-pass filtered at 200 Hz prior to averaging and averaged over 15 activations at each location and orientation. Fifteen trials were used to generate an evoked field, because this number of trials was found to generate a similar noise envelope to the experimental data. Dipole fitting and localization error assessment proceeded as described above. 

## 3. Results

Table 1 lists the DC and power line fields outside and inside the shielded cylinder along each cardinal axis and for the vector magnitude $\left|\vec{B}\right|$, as well as the associated change in the vector magnitude. Figure 2 shows the amplitude spectra for the same locations and orientations. Outside of the shield, the DC field vector magnitude ranged from 35–45 μT, and the 60 Hz field vector magnitude ranged from 270–273 nT/√Hz. At the OPM helmet, the DC field vector magnitude was 50.0 nT and the 60 Hz field vector magnitude was 17.5 pT/√Hz. The mean difference was 58.2 dB and 84.2 dB for DC and 60 Hz, respectively. Shielding seems more effective for power line noise as compared to the DC field, although this could be an overestimate due to the possibility of 60 Hz signals being transmitted directly to the fluxgate recording path. Outside of the shield, the amplitude spectra show a relatively equal distribution of power across the frequency range (i.e., “flat spectrum”), with the exception of power line peaks, several peaks at frequencies unrelated to power line (e.g., 20 Hz), and contributions below 1 Hz. The median amplitude in the 10–200 Hz range is 59.1, 59.3, and 64.8 pT/√Hz for each axis and 108.0 pT/√Hz for the vector magnitude. Inside the shield, we also observe many peaks in the spectrum in the 15–150 Hz range, in addition to 60 Hz. The frequency range of these peaks suggests that they may be related to mechanical vibration [24]. It is also worth noting that the amplitude increases at frequencies below 10 Hz, in contrast to the 1–2 Hz knee outside of the shield. The median amplitude in the 10–200 Hz range is 43, 37, 33, and 71 fT/√Hz for each axis and the vector magnitude, respectively. 

Magnetic field vector mapping within a 24 cm sided cube around the sensor array position reveals an anisotropic environment for MEG measurements. Figure 3 shows the magnetic field components for each Euclidean orientation, as a function of position along each direction. Magnetic field components vary by as much as 30 nT over the measurement distance. While trends along the z-axis are predominantly linear, higher-order spatial trends are evident in other directions. Figure 4 provides a visualization of the field gradients by subtracting the mean field vector across all space from each grid location and presenting the residual as a 3D vector map. Table 2 reports the spherical harmonic coefficients up to fourth order. Mean and standard deviation are calculated by aggregating across field maps recorded over several months (May to December 2023, no degaussing). There are higher-order coefficients with mean values larger than the standard deviation, indicating stable anisotropies in the field encompassing the sensor array. At the fourth order, five of nine coefficients have standard deviation greater than the mean. Given the high mean values, this indicates high variability due to model overfitting at the fourth order. Thus, the vector field map in the volume of interest can likely be well explained by a spherical harmonic expansion up to the third order. 

The Pearson correlation coefficient between each dataset and data predicted from its model was 0.97 ± 0.02, indicating an excellent fit to the data. The correlation between each dataset and the data predicted from the mean model was 0.89 ± 0.03, indicating that the model fit is relatively stable over time. The correlation between the spherical harmonic model fit and unseen data was 0.965, as compared to 0.968 for the training data. This indicates that the coefficients are accurate in estimating the field at new positions within the volume of interest.

Evoked field butterfly plots and topographies for four current dipoles on the phantom arc (azimuthal angle at 60 degrees) are shown in Figure 5. In all cases, a single focal source is the most likely generator of the topography. Results of the dipole fit for all sources, as well as localization errors along each axis, are shown in Figure 6, for experimental and simulated data. Experimentally, the mean error in localization is 0.9 ± 2.9 mm, 0.8 ± 2.9 mm, and 1.3 ± 1.9 mm along the three Euclidean axes, and 4.2 ± 2.2 mm in terms of vector magnitude. The experimentally determined current dipole magnitude was 16 ± 2 nAm with a goodness of fit of 85 ± 7%. We quantify the bias in localization as a t-statistic along each axis, indicating the number of standard deviations between the mean localization error and zero. The variability along each axis is larger than the localization error, indicating that there is no significant bias in localization. We note that localization errors increase substantially for sources 11 and 12, for which the magnetic field is incomplete due to the lack of sensor coverage over the face. Excluding these two outlying sources, the mean error in localization is 0.02 ± 2.2 mm, −0.05 ± 1.9 mm, and 1.7 ± 1.2 mm along the three Euclidean axes, and 3.3 ± 0.8 mm in terms of vector magnitude. After excluding outliers, the variability is substantially reduced. There is no significant bias along the *x*- and *y*-axes, and minimal bias (1.4 standard deviations) in localization along the *z*-axis. The current dipole magnitude and goodness of fit were unchanged within error. The error determined experimentally is about 2.5 times larger than simulation studies with similar current dipole and noise characteristics. 

To test the impact of noise reduction methods on source estimation, we re-calculated the evoked field data without HFC and RAR applied and projected these uncleaned data to the same current dipole location and orientation. Across all twelve phantom sources, there was no evidence for a change in the estimated current dipole magnitude due to HFC and RAR. Noise reduction did not significantly improve the localization error for phantom sources, likely because the data were already filtered to a narrow band (excluding most artefacts). However, a substantial reduction in the amplitude of artefactual peaks in the wide-band spectrum can be observed in Figure 7a–c. Attenuation of artefactual signal power via reference array regression and homogenous field compensation ranged between 0 and 25 dB per channel (5–15 dB on average), indicating that these approaches to noise reduction are effective. Even prior to applying noise reduction, Figure 1e shows that the low-frequency signal drift (0–30 Hz) over three minutes is less than 100 pT. 

## 4. Discussion

The passive shield cylinder provides sufficient attenuation of the DC environmental field to allow for recordings using the OPM sensors supplied by Fieldline, which require an ambient field magnitude less than 100 nT to start operating. In our environment, field fluctuations over time are on the order of 10s of pT, which is well below the 30 nT dynamic range of these sensors. OPM sensors supplied by some other vendors have smaller operating and dynamic ranges. For example, QuSpin sensors require a background field less than 50 nT and have a 5 nT dynamic range. These sensors may function inside of the cylindrical shield in our environment, although there may be a need for active magnetic field nulling coils [6]. Our field mapping process identified substantial anisotropy in the remnant field in the volume encompassing the OPM sensor array. We observed DC field fluctuations of 15–30 nT over a 24 cm distance, providing a rough approximation of the remnant field gradient on the order of 0.6–1.2 nT/cm. In such an environment, even very small sensor movements with respect to the remnant field have the potential to generate interference signals much larger than the neuromagnetic signals of interest. Given the growing interest in performing OPM MEG recordings where head movement occurs (e.g., in children), mitigating field anisotropy in the cylindrical shield will be an essential future step. Spherical harmonic expansion clarified that the remnant field is quite stable over time and generalizes to unmodelled locations. Thus, the coefficients can be used to develop custom-designed field nulling coils for the shielded environment.

While roughly 37 fT/√Hz noise is achieved along the sensor axis with the passive shielding at most frequencies, we found that OPM recordings inside of the shield demonstrated increased signal power at specific frequencies in the 15–150 Hz range. These peaks must be addressed because the increased noise limits sensitivity in critical frequency domains for brain electrophysiology. The frequency characteristics of the artefact suggest a mechanical origin. We propose that relative motion between the sensors and the remnant field are a likely source. This may be due to sensor vibration in a stationary remnant field or to movement of the remnant field caused by vibration in the mu-metal shielding. Similar peaks were observed in the recordings outside of the shield and by other authors [25], suggesting that some noise peaks observed in the OPM recordings may be environmental in nature, or that field fluctuations due to shielding vibration are also captured outside of the shield. Limes et al. also reported a slightly lower unshielded amplitude spectrum above 100 Hz [25], suggesting a potential overestimation of the field in this frequency range, which is not generally relevant for neuromagnetic recordings. A comprehensive assessment of the sources of peaks observed outside of the shield is well beyond the scope of this manuscript. It is worth noting that empty-shield OPM sensor amplitude spectra from other sites show similar patterns [24,26] and that these patterns are not apparent in cryogenic MEG gradiometer recordings where any change in field induced via movement through a spatially flat remnant field would be rejected. This highlights a common problem which has emerged with the switch to OPM MEG; the use of magnetometers means intrinsic noise rejection cannot be achieved, hence the many developments of active field nulling coils to produce a zero field environment [9,27]. 

Noise reduction methods following data acquisition such as reference array regression (RAR), homogenous field correction (HFC), independent component analysis (ICA), signal space separation (SSS) [28,29] and adaptive multipole modeling (AMM) [30] are potential data-driven mechanisms to mitigate the impact of artefactual peaks in the OPM amplitude spectrum. Here, we implemented RAR and HFC to provide a 5–15 dB reduction in noise during phantom recordings. In MEG, ICA is most commonly used to remove biomagnetic artefacts such as heart beats and eye blinks, which would not occur in our phantom data. However, ICA could be applied in addition to HFC and RAR for human recordings. Both SSS and AMM require more concurrently recorded sensors than were available in this study for effective use. As such, these methods were not tested here. Importantly, noise reduction with HFC and RAR did not change the estimated current dipole magnitude, indicating that a substantial SNR boost is achieved with these methods. Importantly, the shielding factor achieved here is comparable to values reported for 86- and 128-channel systems where data are recorded concurrently [24,30]. This speaks to the viability of using consecutive recordings to achieve high-density whole-head OPM data.

We found roughly 3 mm localization error in experimental recordings of tangential current dipoles at a radius of 6.5 cm from the origin, with no evidence of substantial bias. While greater than simulated localization error, the variability and bias reported here are in line with previous reports using previously validated phantoms in OPM and cryogenic MEG, as follows. A similar dry phantom design resulted in 1–2 mm localization error when used in a whole-head SQUID MEG system, although this accuracy was achieved after re-calibration of the phantom [16]. In OPM MEG, similar or better localization accuracy has been reported for a dry phantom; however, dipole magnitudes in this case were on the order of 1 μAm, which is two orders of magnitude larger than brain activity [9,31]. A phantom localization error of 5 mm was found for current flowing over a 1 cm distance inside of a saline solution sphere using OPM MEG sensors [5]. Given the novelty of the phantom used in this study, our results validate the accuracy of both the phantom and the OPM system (i.e., vendor-supplied sensors and helmet). We recognize that misplacement of the phantom platform in the helmet could lead to bias in localization, which is a design limitation. A solution for future consideration is installing small wire loops at known positions on the phantom, which can be activated to generate magnetic dipoles. These magnetic dipoles can act as fiducial markers (i.e., head-position-indicator coils) that can be localized to precisely register the phantom’s location within the helmet. Importantly, this will increase the ecological validity of our phantom by mimicking a common operating procedure for human MEG recordings. 

In conclusion, we have shown that OPM recordings work well inside of a shielded cylinder in our environment. High-density whole-head recordings can be achieved by combining consecutive recordings with appropriate noise reduction techniques. We expect that increasing the number of concurrently recorded sensors and introducing field nulling will further improve data quality and allow sensor movement. In the long run, high-quality recordings on a portable and cheaper platform will improve access to MEG worldwide.

## Figures and Tables

**Figure 1 sensors-24-03503-f001:**
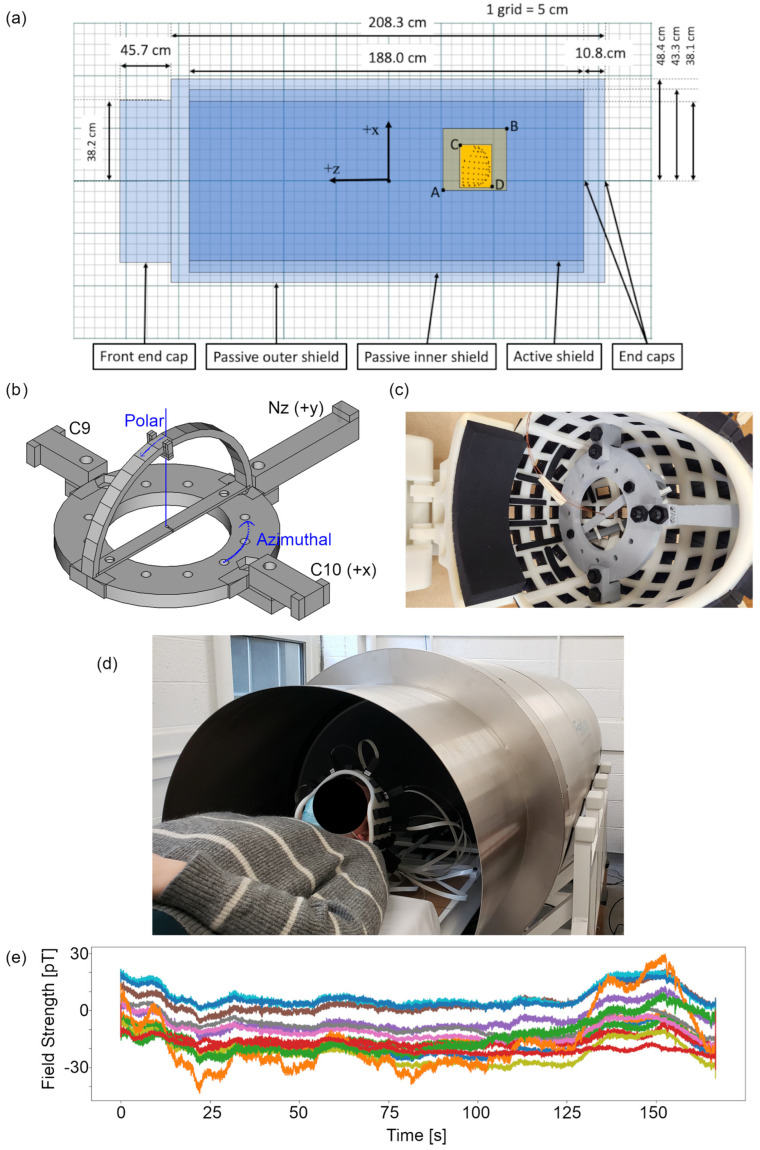
System coordinates and phantom design. (**a**) The 2 passive and 1 active shielding cylinders are shown as semi-transparent blue rectangles. Dimensions for the cylinders and end caps are provided in the cylinder coordinate system. The location of the helmet and the volume for field mapping are shown as overlapping semi-transparent orange rectangles. Points A and B indicate a 24 cm sided cube within which the vector magnetic field will be mapped. Points C and D indicate the volume containing the sensor array (i.e., helmet). (**b**) Schematic of the phantom with the relevant angles for the coordinate system indicated in blue. (**c**) Phantom mounted in the OPM helmet. (**d**) Participant ready to be inserted into the shield. (**e**) OPM time courses from a single recording. Each line represents the magnetic field recorded at one of sixteen sensors (one colour per sensor) during a phantom recording. A 30 Hz low-pass filter (no high pass) was applied to the data to highlight raw signals in the lower-frequency regime.

**Figure 2 sensors-24-03503-f002:**
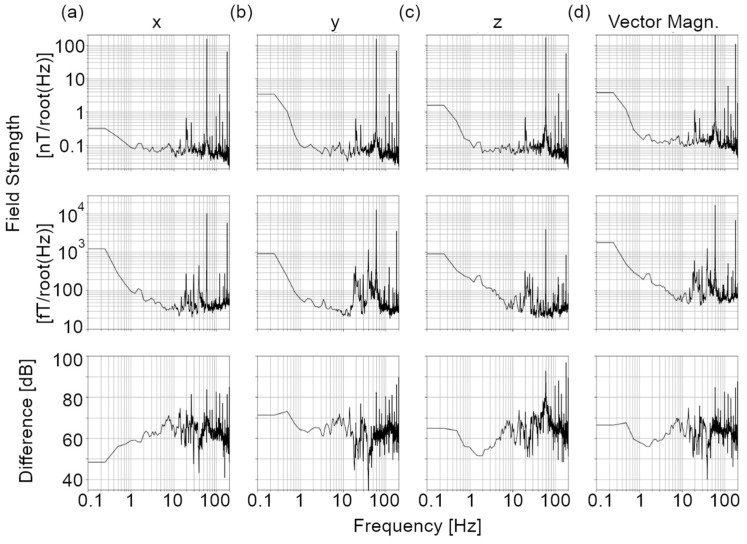
Magnetic field and shielding factor spectra. Magnetic field spectra acquired (top row) outside of the shield and (middle row) at the centre of the helmet are shown, as well as the (bottom row) associated shielding factor, for sensors oriented (**a**) posterior to anterior, (**b**) left to right, and (**c**) superior to inferior. Coordinates are with respect to a human participant in a supine position with their head in the helmet. (**d**) The vector magnitude spectra and associated shielding factors.

**Figure 3 sensors-24-03503-f003:**
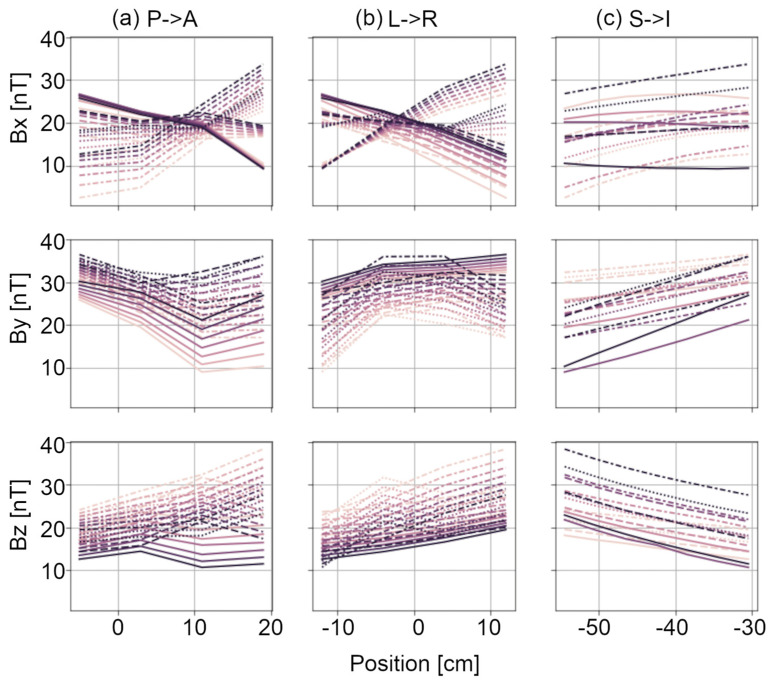
Vector field within the volume of interest. DC field vector strength along each cardinal axis is provided as a function of location within a volume encompassing the OPM helmet. Field vector amplitudes in the (top row) x, (middle row) y, and (bottom row) z orientations are shown with respect to position in the (**a**) x, (**b**) y, and (**c**) z directions. Data are shown for points on a 4 × 4 × 8 (x-y-z) grid. Each line in each plot represents field vector amplitude along one line.

**Figure 4 sensors-24-03503-f004:**
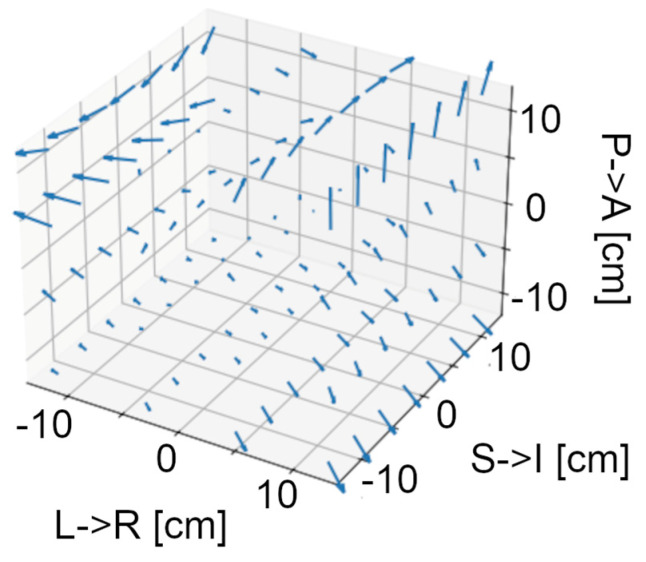
A 3-D vector representation of the field map. The arrows represent the field vector at each point in space, with orientation accurately represented and arrow length proportional to field strength. The mean field across the volume is subtracted from each vector to highlight the field gradients across the volume.

**Figure 5 sensors-24-03503-f005:**
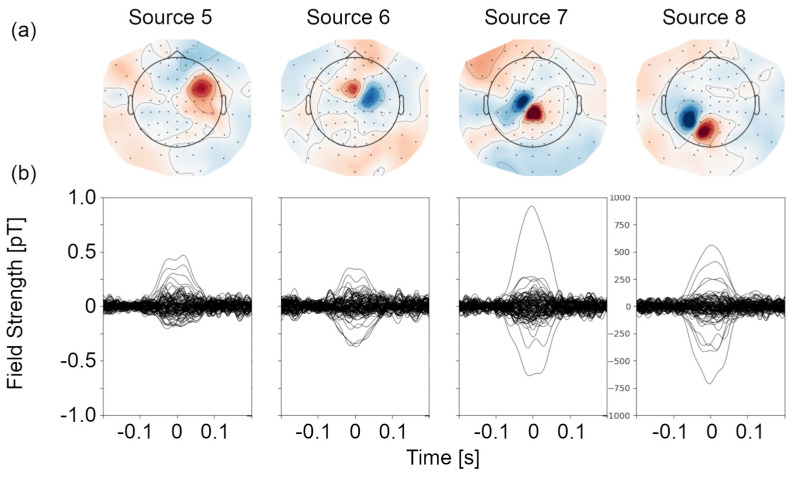
Evoked field data for phantom sources. (**a**) Evoked field topographies generated by activation of four current sources on the phantom. A schematic of the helmet is superimposed in each topography to clarify the spatial arrangement of sensors. Participant left is on the left and the nose is at the top for each topography. (**b**) Evoked field as a function of time for all 104 OPM sites and four current sources on the phantom.

**Figure 6 sensors-24-03503-f006:**
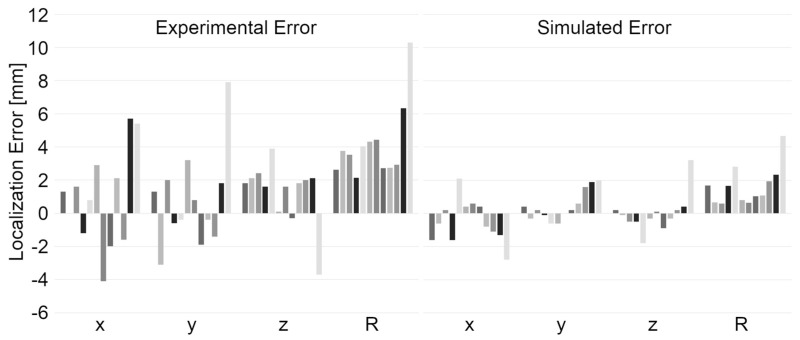
Localization errors (LE) for measured and simulated equivalent current dipoles. LE is shown for each of the 12 sources, along each cardinal axis (x, y, z) and as a vector magnitude (R). LE measured with a phantom in the OPM system is shown on the left. LE measured based on simulations with the same parameters is shown on the right. Each coloured column indicates LE for a different phantom source location.

**Figure 7 sensors-24-03503-f007:**
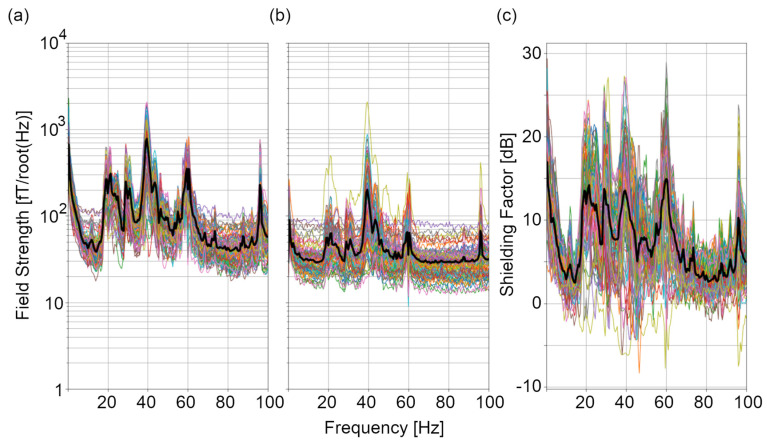
Noise reduction in phantom recordings. Magnetic field spectra (**a**) prior to and (**b**) following reference array regression and homogenous field correction for all 104 OPM recording sites. (**c**) The shielding factor spectra associated with these two processes are also shown. Black lines in (**a**–**c**) indicate the mean across all sensors. Each coloured lines indicates the spectrum for one recording site in the helmet.

**Table 1 sensors-24-03503-t001:** Environmental magnetic fields and shielding factors.

	DC				60 Hz			
	B_x_	B_y_	B_z_	$\left|\vec{B}\right|$	B_x_	B_y_	B_z_	$\left|\vec{B}\right|$
Outside Shield	μT				nT/√Hz			
Location 1	−27.5	19.4	29.9	45.0	155.0	154.7	161.1	271.9
Location 2	−34.3	−5.92	17	38.8	152.1	153.7	167.4	273.5
Location 3	−31.7	25.3	18.9	44.7	148.5	148.1	171.2	270.1
Location 4	−25.4	3.63	24.9	35.8	151.7	149.2	168.4	271.3
Inside Shield	nT				pT/√Hz			
Helmet centre	−28.3	−38.6	−14.8	50.0	10.2	12.6	3.91	16.7
**Mean Difference [dB]**								
				58.2				84.2

**Table 2 sensors-24-03503-t002:** Spherical harmonic components: means and standard deviations.

		*m*								
	*l*	−4	−3	−2	−1	0	1	2	3	4
Mean	1 [nT]				18.6	26.5	21.9			
	2 [nT/m]			−19.5	25.6	36.9	−18.8	−3.29		
	3 [nT/m^2^]		112	12.6	155	0.127	4.17	−13.3	9.20	
	4 [nT/m^3^]	944	42.4	−1.38	11.3	−4.23	37.4	−32.1	32.1	7.22
Std. Dev.	1 [nT]				4.98	4.74	2.40			
	2 [nT/m]			12.2	8.20	4.90	6.25	5.80		
	3 [nT/m^2^]		16.0	4.86	41.6	5.85	3.49	6.02	2.53	
	4 [nT/m^3^]	165	63.8	8.46	48.2	4.83	12.6	15.1	47.3	27.0

## Data Availability

Data are available from the corresponding author upon request via email.

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
