# Peer review of "Noise Reduction and Localization Accuracy in a Mobile Magnetoencephalography System"

_sensors, 2024, doi:10.3390/s24113503_

Round 1

Reviewer 1 Report

Comments and Suggestions for Authors

Summary

Magnetoencephalography (MEG) non-invasively measures the weak magnetic fields generated by the human brain. Most MEG systems rely on cryogenic sensors housed inside a dewar and require a large shielded room. The use of optically pumped magnetometers (OPMs) instead of cryogenic sensors makes person-sized magnetic shields possible. In this article the performance of OPM recordings inside a person-sized cylindrical shield is investigated. First, the efficacy of the shield is determined and the distribution of the remnant magnetic field in the relevant sensor volume measured. The recordings show that the shield provides sufficient attenuation to operate the OPMs. Second, the localization accuracy of known magnetic field sources measured with the OPM sensors was determined and found to be around 3 mm, similar to previous reports. The results demonstrate the viability of using OPM magnetometers in a smaller, cheaper, and more mobile magnetic shield for MEG measurements. The paper is valuable as a benchmark for other groups building similar systems and a step towards making MEG more accessible worldwide.

List of comments

1.       Using a person-sized human shield for MEG was already tested in 2006 and described in this paper: H. Xia et al.: "Magnetoencephalography with an atomic magnetometer," Applied Physics Letters, vol. 89, no. 21, p. 211104, 2006. This paper should definitely be added as a reference, for example in line 72.

2.       A photograph of the shield would make it easier for a person not working in the field to imagine how the system looks like and give a feeling for the size of the shield. I highly recommend adding such a photograph to figure 1.

3.       Table 1 contains the coordinates of points A-D shown in figure 1a. These coordinates are not really relevant and it would make sense to just add them to figure 1a. Also, in the caption of figure 1a, is says “points A-B” instead of A-D.

4.       Table 2 shows the magnetic field at four locations outside the shield. While the field magnitudes are fairly similar, the orientation of the field changes a lot, resulting in strong variations in the three vector components. To calculate the shielding factor at a certain point, the magnetic field at this point with and without the shield needs to be known. Using the mean magnitude of the field outside the shield to calculate the shielding factor makes sense. Using the mean for each vector component, however, is not reasonable as the components are so different and the mean does not necessarily give a good estimate of the magnetic field component at the place of the helmet without the shield. Since the shield is mobile, one could actually measure the magnetic field at the helmet position with the shield removed. Furthermore, the whole calculation only makes sense if the magnetic field outside the shield stays more or less constant over time (both magnitude and orientation). For the paper, I suggest only calculating the shielding factor from the magnitude and removing the shielding factors for the individual components.

5.       Table 2 would be easier to read if there was a vertical line between the DC and the 60 Hz part. Also, x, y, and z should be replaced by Bx, By, Bz as the values are not coordinates. The vector magnitude should be replaced by |B| (with a vector on the B) or the caption should say, what R is. In line 295, where R is defined, it says “vector sum”. This should be the vector magnitude instead. Here one could add R or |B|.

6.       In line 305 and 306, it says that the mean amplitude in the 10-200 Hz range is 1.1 nT/root(Hz) for each axis and 2.0 nT/root(Hz) for the vector magnitude. In Figure 2, the two values seem to be about 10 times higher, i.e. 11 nT/root(Hz) and 20 nT/root(Hz).

7.       In line 314, the attenuation is estimated as 90 dB, however, in figure 2 it looks more like 80-90 dB, which is also what is listed in the abstract. Please adjust this.

8.       In figure 2, the notation is suddenly changed from the x-y-z coordinates also used in figure 1a to “posterior to anterior” shortened to “P ->A” and similar. This is confusing and makes me loose orientation. Please also change to the x-y-z notation in figures 2 and 3.

9.       In figure 2, the limited bandwidth of the flux gate and the OPMs is clearly visible in form of a roll-off at high frequencies. This results in an unreal high shielding factor above 200 Hz. I suggest only showing the data up to 200 Hz.

10.   In figure 2, the separate shielding factors for the three vector components are more reasonable than in table 2 since the outside magnetic field components look very similar. (This is just a comment and no changes are needed.)

11.   In table 3, I would add to the caption what the bold formatting of the values means.

12.   In line 373/374, you write “The variability along each axis is larger than the localization error, indicating that there is no significant bias in localization.” Once you exclude the two outliers, this is no longer the case for the z direction. How do you justify your reasoning? Also, how is the bias calculated?

13.   Table 4 would profit from value-dependent background color to more easily see big/small values. A graph instead of a table could also be an option. Currently, the information is drowned in the large number of entries in the table.

14.   In line 394, the low frequency signal drift is claimed to be “on the order of 10-30 pT”, however, in figure 6d, the orange trace varies by roughly 70 pT. I would therefore calculate the mean and the standard deviation of the drift and state these values instead.

15.   In line 425, the 90 dB should again be changed to 80-90 dB.

16.   In the sentence in line 436-439, references should be added.

17.   In the supplemental Table 1, I find the notation unclear. Is the differentiation between magnetic field components and coordinates done correctly? For the l=2 components, what is meant with the part in the bracket with the equal sign?

Small language/formatting errors:

18.   This sentence in line 51-53 does not make sense: “Recent advances in the miniaturization of quantum sensors have seen the increased use of optically pumped magnetometer (OPM) technology has recently been adapted for MEG recordings [5,6].” Please adjust the wording.

19.   In the sentence in lines 170-173, a “maximal” should be added: “… with maximal field gradients on the order of …”. Else it sounds like lower values do not provide the listed benefits.

20.   In the caption of figure 5, line 355, an “and” or similar is missing.

21.   In line 362, a “for” is too much.

22.   In line 465, it is unclear what “1-cm” means.

23.   Several units are formatted wrongly and should be corrected:

a.       “root” instead of the square root symbol: line 126, line 136, twice in table 2, line 298, line 299, line 305, line 306, line 312, twice in figure 2.

b.       “u” instead of µ for micro, like in uAm and uT: line 141, line 297, line 464.

c.       cm3 in line 182.

d.       Line 219/220: degree turned into a 0, which is very confusing. Same in line 367.

e.       Line 223: 500 kΩ and 5 µA.

f.        Line 225: use 1000 Hz instead of 1,000 Hz like everywhere else.

g.       The +/- could be replaced by the correct symbol.

24.   The Data Availability Statement is missing (lines 490-495).

25.   The references need checking for completeness.

Reviewer 2 Report

Comments and Suggestions for Authors

I was glad to see variations magnetic field strength inside the screen. It is a pity that there is no recording of the geomagnetic field outside the screen to assess the origin of the variations.

Reviewer 3 Report

Comments and Suggestions for Authors

In this paper, the authors proposed a small cylindrical magnetically shielded room for magnetoencephalography. The resulting shielding factor obtained passive shielding and additional signal processing satisfies the operational requirements of the OPMs and is sufficient to obtain a magnetic field environment in which MEG signals can be detected. In this regard, this paper is considered to be of interest to many OPM users and worthy of publication.

However, there are several points of concern including the method for calculating the shielding factor. Please revise the paper before publication, taking into account the following comments.

1. The authors estimated the shielding factor as 90 dB from the noise in the 10-200 range. However, "flat spectrum" in Figure 2 indicates that the magnetic field signals measured inside and outside the shielded room are uncorrelated, and therefore, taking the ratio of these noise levels to determine the shielding factor is meaningless. The authors has merely calculated the ratio of noise in the 10-200 range between OPM measurements and the fluxgate measurements, rather than evaluating the performance of the shielded room.

2. The authors estimated the shielding factor at 60 Hz by taking the peak amplitude from Figure 2. However, the unit of the peak values is given as noise density (T/sqrt(Hz)). These values were obtained by normalizing the amplitude spectrum obtained by FFT using sqrt(N/f), where N is the number of data points and and F is the sampling rate. Therefore, if one were to determine the shielding factor from the ratio of these peak values, it would be necessary to use the peak values of the amplitude spectrum before normalization, or to align the sqrt(N/f) for both OPM and fluxgate. Has this been considered?

3. Anyway, the estimation of 60-Hz shielding factor using the peak amplitude of Figure 2 was inappropriate as it likely includes not only the magnetic field noise detected by the flux sensors but also noise is electrically mixed in without passing through the sensors. To measure the frequency characteristics of the shielding factor, rather than using such passive methods, it is necessary to prepare a large coil and actively apply magnetic fields of various frequencies to determine the extent to which they are shielded through the sensors.

4. The authors estimated the shielding factor along each cardinal axis, but normally, the direction of the magnetic field inside and outside a shielded room is distorted by the shielded room itself. This is especially pronounced when the shielded room is cylindrical. Therefore, instead of producing a shielding factor for each axis, how about discussing only the vector sum (R)? Or, consider dividing the shielding factor into components along the z-axis and the orthogonal components in the X-Y plane (sqrt(X^2 + Y^2)) based on symmetry.

5. Unless there is a specific reason, the author should align the labels P->A, L->R, S->I in Figures 2-4 with X, Y, Z respectively. It should be more understandable for readers.

6. I could not understand what "azimuthal angle at 600" on L367 meant. Was it a typo for "azimuthal angle at 60 deg"? If so, why the azimuthal angle for the source 5-8 shown in Fig. 5(a) were 60 and 240, not all 60? Since the difference between 60 deg and 240 deg is 180 deg, does this mean it indicates a phase reversal? It's confusing for me.

7. Related to the comment #6, what are the units of the angle values (650, 250, -150, -550, 600, 1200) on L219 and L220? Do these mean (65, 25, -15, 55, 60, 120) degree? If possible, additionally illustrating the angles in Fig. 1(a) would clarify their consistency with the magnetic field pattern in Fig. 5(a). 

8. On L223, correct the units for 500 and 5, as part of them is garbled. 

9. Please provide references for the part 'While...OPM and cryogenic MEG' on line 457.

10. On L305-306, should "1.1 nT/root(Hz)" and "2.0 nT/root(Hz)" be corrected to "11 nT/root(Hz)" and "20 nT/root(Hz)" according to Figure 2?

Reviewer 4 Report

Comments and Suggestions for Authors

Dear Authors, 

Thank you for the interesting article. I have only a couple of minor comments, which might be considered to be questions. In my opinion, the article does not demand a revision. I highly recommend the article for publication as is.

Minor comments (questions to authors): 

1. Lines 124-125: how sliding into slots in the helmet affects orientations? Are orientations be hardly fixed, or can they be changed?

2. Lines 156-157. From my experience, Quspin OPMs produce artifacts at frequency 77 Hz when using the frequency sampling of 1kHz. Do Fieldline magnetometers produce these parasite peaks, or does this problem related to Quspin sensors only?

Round 2

Reviewer 3 Report

Comments and Suggestions for Authors

As I pointed out in my first review, using the ratio of noise floor values in the 10-200 Hz range as a shielding factor is inappropriate because there is no correlation between the noises of the two. It merely represents the ratio of the noise from the magnetic sensors and the measurement system, not the shielding factor. Therefore, the plots in the bottom row of Figure 2 are meaningless. The authors should have realized this when they changed the data acquisition unit from NI USB-6009 to NI USB-6216, as the values changed. If the noise could also be measured outside with an OPM, the noise floor in the 10-200 Hz range would differ from that in the top row, and thus the plots in this bottom row would change.

Principally, to clearly determine the shielding factor, it is necessary to apply a magnetic field while scanning frequencies using a coil from outside, measure the applied magnetic field signals inside and outside, and take their ratio. However, the authors insist that such coil study is beyond the scope of this manuscript. To give ground, if the authors choose to discuss the shielding factor based on the top row and middle row of Figure 2, the discussion should be limited only focusing on the peaks that are common to both (for example, 20 Hz, 26 Hz, 40 Hz, 60 Hz, 120 Hz, 180 Hz).
